# p53-Mediated Radiosensitization of ^177^Lu-DOTATATE in Neuroblastoma Tumor Spheroids

**DOI:** 10.3390/biom11111695

**Published:** 2021-11-15

**Authors:** Sara Lundsten, Hanna Berglund, Preeti Jha, Cecilia Krona, Mehran Hariri, Sven Nelander, David P. Lane, Marika Nestor

**Affiliations:** 1Department of Immunology, Genetics and Pathology, Uppsala University, SE-751 85 Uppsala, Sweden; sara.lundsten@igp.uu.se (S.L.); hanna.berglund@igp.uu.se (H.B.); preeti.jha@igp.uu.se (P.J.); cecilia.krona@igp.uu.se (C.K.); mehran.hariri@igp.uu.se (M.H.); sven.nelander@igp.uu.se (S.N.); dplane@imcb.a-star.edu.sg (D.P.L.); 2Department of Medicinal Chemistry, Uppsala University, SE-751 23 Uppsala, Sweden; 3p53Lab, Agency for Science Technology and Research (A*STAR), Singapore 138648, Singapore; 4Department of Microbiology, Tumor and Cell Biology, Karolinska Institute, SE-171 65 Solna, Sweden

**Keywords:** neuroblastoma, radionuclide therapy, p53, ^177^Lu-DOTATATE, radiosensitization, stapled peptides

## Abstract

p53 is involved in DNA damage response and is an exciting target for radiosensitization in cancer. Targeted radionuclide therapy against somatostatin receptors with ^177^Lu-DOTATATE is currently being explored as a treatment for neuroblastoma. The aim of this study was to investigate the novel p53-stabilizing peptide VIP116 in neuroblastoma, both as monotherapy and together with ^177^Lu-DOTATATE. Five neuroblastoma cell lines, including two patient-derived xenograft (PDX) lines, were characterized in monolayer cultures. Four out of five were positive for ^177^Lu-DOTATATE uptake. IC_50_ values after VIP116 treatments correlated with p53 status, ranging between 2.8–238.2 μM. IMR-32 and PDX lines LU-NB-1 and LU-NB-2 were then cultured as multicellular tumor spheroids and treated with ^177^Lu-DOTATATE and/or VIP116. Spheroid growth was inhibited in all spheroid models for all treatment modalities. The most pronounced effects were observed for combination treatments, mediating synergistic effects in the IMR-32 model. VIP116 and combination treatment increased p53 levels with subsequent induction of p21, Bax and cleaved caspase 3. Combination treatment resulted in a 14-fold and 1.6-fold induction of MDM2 in LU-NB-2 and IMR-32 spheroids, respectively. This, together with differential MYCN signaling, may explain the varying degree of synergy. In conclusion, VIP116 inhibited neuroblastoma cell growth, potentiated ^177^Lu-DOTATATE treatment and could, therefore, be a feasible treatment option for neuroblastoma.

## 1. Introduction

The transcription factor p53 is a tumor suppressor involved in DNA damage response and induces cell cycle arrest or cell death in damaged cells via several pathways [1]. Utilizing the p53 signaling pathway for potentiation of radiotherapy in cancer is therefore a promising approach to improve therapeutic outcomes. The fact that nearly all tumors have inactivated the p53 pathway demonstrates the importance of this pathway in tumorigenesis [2]. Inactivation of p53 can for example be achieved by mutation in the p53 gene, overexpression of negative regulators such as MDM2 and WIP1 or inactivation of downstream targets such as p21 (WAF1) [1,3,4].

For wildtype p53 (wtp53) tumors, inhibition of the p53-MDM2 interaction is emerging as an approach to stabilize p53, and several molecules are currently in clinical trials [5,6]. One problem with MDM2-targeted inhibition is resistance due to overexpression of the homologous MDM4 (also called MDMX) protein [5,7]. As a consequence, the development of molecules targeting both of these p53-inhibiting proteins is of importance to reach clinical success.

The small stapled peptide VIP116 and its predecessor PM2 (sMTide-02) have high affinity for MDM2 and MDM4 [8,9]. We have previously demonstrated promising therapeutic effects of PM2 and VIP116 in a variety of wtp53 tumors in vitro [10,11]. Moreover, PM2 was able to potentiate radiotherapy in a preclinical setting [12,13,14].

Using stapled peptides for inhibition of the MDM2/MDM4-p53 interaction is a relatively new concept, although encouraging results from a phase I clinical trial, with once- or twice-weekly dosing, demonstrates the feasibility and tolerability of these compounds [15].

Targeted radionuclide therapy against the somatostatin receptors (SSTR) with ^177^Ludodecane tetraacetic acid (DOTA)-Tyr3-octreotate (^177^Lu-DOTATATE, Lutathera) has recently reached clinical approval after over 25 years of development [16,17]. One major benefit this type of targeted radiotherapy is the cross-fire effect. This concept, where the radiation is delivered to not only target-expressing tumor cells but also adjacent cells, can help overcome resistance due to low receptor density [18]. ^177^Lu-DOTATATE is approved for therapy of gastroenteropancreatic neuroendocrine tumors, although ongoing research is currently identifying other types of tumors which could benefit from the therapy. Among these are neuroblastoma, a pediatric tumor arising from the sympathetic nervous system. In this disease group, the high-risk cases often demonstrate metastatic disease with high resistance against therapy [19].

Several studies have studied SSTR expression in neuroblastoma, and the number of SSTR2 (to which of the five receptors ^177^Lu-DOTATATE has the highest affinity) positive neuroblastoma tumors is reported to range from 75 to 90% [20,21,22,23,24]. Early phase clinical trials have been conducted with ^177^Lu-DOTATATE or similar derivatives in neuroblastoma with promising results [25,26,27,28]. Moreover, several clinical trials further exploring radionuclide therapy against SSTR in neuroblastoma are currently being conducted (NCT03273712, NCT02441088, NCT03966651, NCT04903899, NCT04023331) [6].

As neuroblastoma has very low rates of p53 mutations, there is great potential of utilizing p53 stabilization as a therapeutic option, both as monotherapy and together with targeted radionuclide therapy. A number of p53-MDM2 inhibitors have been tested preclinically in neuroblastoma with positive results [29,30,31,32,33,34,35,36]. HDM201, a compound from Novartis, is currently in phase 1 trials for treatment of neuroblastoma (NCT02780128) [6]. One preclinical study has explored the use of MDM2-inhibitor Nutlin-3 as potentiator of ^177^Lu-DOTATATE with encouraging results [37].

The aim of the present study is to investigate the potency of the novel p53 stabilizing peptide VIP116 and ^177^Lu-DOTATATE, both as monotherapies and in combination, in multicellular neuroblastoma tumor spheroids. The study is performed both in commercially available cell lines and by the use of patient-derived xenograft (PDX)-models established from neuroblastoma patients.

## 2. Materials and Methods

### 2.1. Statistical Analysis

All experiments included in the study were performed in a minimum of two independent replicates. Data is presented as pooled data from all replicates or from one representative assay. Statistical analysis was performed with Prism 9 (Graphpad, CA) unless otherwise stated. Statistical significance is presented as not significant (n.s., *p* ≥ 0.05), * (*p* < 0.05), ** (*p* < 0.01), *** (*p* < 0.001) and **** (*p* < 0.0001).

### 2.2. Cell Lines

All cells were cultured according to manufacturer’s instructions at 37 °C with 5% CO_2_. The human adenocarcinoma cell line HCT116 (with wt p53) and the corresponding p53 knock-out cell line (HCT116 p53-/-) were obtained from Horizon discovery (Cambridge, UK). The human neuroblastoma cell lines IMR-32 and SK-N-AS (SKNAS) were purchased from American Type Culture Collection(Manassas, VA, USA). The murine neuroblastoma cell line Neuro2a (N2a) was purchased from Sigma Aldrich (Darmstadt, Germany). HCT116 and HCT116 p53-/- were cultured in RPMI (Biowest, MO, USA), IMR-32 and N2a were cultured in MEM Earle’s (Biochrom, Berlin, Germany or Sigma Aldrich, Darmstadt, Germany) and SKNAS was cultured in DMEM. Media was supplemented with 10% (HCT116, HCT116 p53-/-, N2a and SKNAS) or 20% (IMR-32) fetal bovine serum (Sigma-Aldrich, Darmstadt, Germany), 1% antibiotics (100 IU penicillin and 100 μg/mL streptomycin) and 1% L-glutamin (Biochrom, Berlin, Germany ). SKNAS and N2a media was further supplemented with 1% non-essential amino acids (Thermo Fisher, Waltham, MA, USA). Cells were passaged one to three times per week with 0.25% trypsin-EDTA (Life Technologies/Thermo Fisher, Waltham, MA, USA).

The PDX cell lines LU-NB-1 (NB1) and LU-NB-2 (NB2) were kindly provided by Prof. Bexell (Lund University, Lund, Sweden) and cultured in DMEM/GlutaMAX^TM^ F-12 (3:1 ratio, both from Gibco-Life technologies/Thermo Fisher, Waltham, MA, USA) supplemented with 1% antibiotics, 2% B-27 w/o vitamin A (Gibco-Life Technologies), 40 ng/mL basic FGF (Peprotech, Stockholm, Sweden) and 20 ng/mL EGF (Peprotech) [38,39]. Cells were grown as neurospheres and passaged one to two times per week using accutase (Sigma Aldrich) with subsequent filtering in a 50 μm cell strainer (Sysmex, Norderstedt, Germany) and re-seeding at a cell concentration of 10^5^ cells/mL.

### 2.3. Drug and Radioconjugate Preparation

^177^Lu-DOTATATE was prepared as previously described [40]. Briefly, DOTATATE (Bachem, Bubendorf, Switzerland) was dissolved in MQ water and aliquots were stored in −20 °C. For labeling, 1.5 µg DOTATATE (Bachem, Bubendorf, Switzerland) was mixed with 60 MBq (Bq = nuclear decays/second) ^177^Lu (ITM, Munich, Germany), approximately 25 µL fresh labeling buffer (25 mM sodium ascorbate/50 mM sodium acetate, pH 5) and heated at 80 °C for 30 min.

To validate the labeling protocol, purity of ^177^Lu-DOTATATE was assessed by HPLC (Phenomenex Luna column, 4.6×100 mm, 3 Å, Phenomex, Vaerlose, Denmark) with the gradient of 5–100% on solvent B: Acetonitrile, 0.1 %TFA (solvent A: MQ-water, 0.1 %TFA; flow rate 1.0 mL/minutes, 10 min, 220 nm). Independent gradients were run for the commercially obtained DOTATATE and on-site radiolabeled ^177^Lu-DOTATATE. For routine labeling, instant layer thin layer chromatography (ITLC, sodium citrate solution, 0.1 M, pH 5.5) was used to validate labeling yield, as previously described [40].

To validate the stability for transmetallation and transchelation, ^177^Lu-DOTATATE (2.8–3.1 µL, 5 MBq) were incubated in 50% fresh mouse serum in PBS (500 µL) or freshly prepared 100 mM EDTA solutions (3000 molar excesses, 500 µL) in MQ-water. Free ^177^Lu (0.6 µL, 5 MBq) with PBS, Saline (0.9% NaCl solution), and 100 mM EDTA solutions were used as controls. The reaction mixtures were incubated at 37 °C for up to 96 h and the respective amount of free ^177^Lu was quantified with ITLC at 0, 24, 48, 72 and 96 h.

For investigation of unspecific effects from free ^177^Lu in spheroid assays, ^177^Lu was coupled to DOTA (Macrocyclics, Plano, TX, USA) by mixing 5 MBq ^177^Lu with 17 µg DOTA dissolved in sodium acetate (0.2 M, pH 5.5) and heated at 80 °C for 30 min. Labeling yield was assessed as described for DOTATATE [40].

VIP116 was produced by the p53 lab and a 10 mM stock solution in DMSO was prepared and stored at −20 °C [8,9]. VIP116 aliquots were prior to use diluted in cell media and added to cells.

### 2.4. XTT Viability Assay

An XTT cell proliferation assay was performed to screen the cell lines for VIP116 sensitivity. In order to culture NB1 and NB2 as adherent cells, plates were coated with 5 μg/mL laminin (LN-521, Biolamina, Sundbyberg, Sweden) according to manufacturer’s instructions at least 24 h prior to seeding. 8000 IMR-32, 1500 N2a, 16000 NB1, 16000 NB2 or 7000 SKNAS cells were seeded in 96-well plates and incubated at 37 °C for 24–48 h. Cell media was removed and replaced by fresh media with 0.1–40 μM VIP116. 72 h after treatment, an XTT assay (301011K, LGC Standards, Middlesex, UK) was performed according to the manufacturer’s protocol. Briefly, XTT activation reagent, XTT reagent and cell media was mixed. Old media was removed and 150 μL of the XTT solution was added to each well. The plates were then incubated for 4 (IMR-32, N2a, SKNAS) or 6 h (NB1, NB2) at 37 °C and analyzed by plate reader (Bio-Rad Laboratories, Hercules, CA, USA). The 50% of maximal inhibition (IC_50_) value was calculated by fitting the data to a dose-response curve (variable slope-normalized response).

### 2.5. Sequencing of the p53 Gene

The p53 status was investigated with targeted Sanger sequencing as previously described [11]. For murine cell lines, all primer sequences except for exon 6, 8 and 9 were kindly provided by Prof. Tweddle (Newcastle University, Newcastle, UK) [41]. Genomic DNA was extracted from frozen cell pellets according to manufacturer’s protocol (DNeasy Blood and Tissue Kit, Qiagen, Germany).

PCR amplification of exon 2-11 was performed with GoTaq G2 DNA Polymerase (Promega, Sweden) by mixing 50 ng DNA with 10 μL reaction buffer, 1 μL of PCR nucleotide mix (10 mmol/L, Promega, Nacka, Sweden), 0.25 μL of GoTaq polymerase, 0.5 μL of primer mix (100 pmol/μL; forward and reverse, produced by Eurofins genomics, Ebersberg, Germany) and H_2_O reaching a final volume of 50 μL. A complete list of all primers used can be found in Appendix A. The reaction was performed in a thermal cycler with the following protocol: 10 min at 95.0 °C; 15 cycles with 30 s at 95.0 °C, 30 s at 64.5 °C, and 30 s at 72.0 °C, followed by 20 cycles with 30 s at 95.0 °C, 30 s at 57.0 °C, and 30 s at 72.0 °C, and a final extension for 5 min at 72.0 °C. PCR products were visualized by use of a 2% agarose gel with TBE buffer.

PCR products were purified and sequenced in forward and/or reverse direction by Eurofins genomics. Sequence alignment was carried out using the SnapGene Viewer 5.3.2 software (GSL Biotech LLC, Chicago, IL, USA) against the human mRNA reference sequence NM_000546_6 and the NC_000017.11 genomic sequence from assembly hg38. The International Agency for Research on Cancer TP53 database (http://p53.iarc.fr/, accessed on 1 November 2021) was used to determine the significance of identified human sequence variants. The mouse derived cell line N2a was aligned against the Trp53 NM011640.3 mouse transcript 1 and the NC_000077 mouse genomic reference sequence from the GRCm39 assembly.

### 2.6. Validation of VIP116 Specificity

To verify the p53-specific activity of VIP116, HCT116 (p53wt) and HCT116 p53-/- cells were used. First, a spheroid assay was performed where spheroids were treated with VIP116 and external beam radiation. HCT116 or HCT116 p53-/- spheroids (1000 cells per well) were prepared and maintained as described in the *Multicellular tumor spheroids* section below. Spheroids were left unirradiated, or irradiated with 2 or 4 Gy (225 kV X-rays at a dose-rate of 1.5 Gy/min using an inherent 0.8 mm Ba filter and an external 0.3 mm Cu filter, X-RAD iR225, Precision X-ray, CT). 3 h later, 0–32 μM VIP116 was added and spheroids were followed for 8 days.

Secondly, a clonogenic survival assay was performed as previously described [12,42]. In brief, HCT116 or HCT116 p53-/- cells were seeded in 6-well plates. Twenty-four hours after seeding, plates were left unirradiated or irradiated with 2 or 4 Gy. Three hours later, media was renewed with or without 16 μM VIP116. Colonies were left to grow for 12 days, after which cells were fixed, stained and counted.

### 2.7. Cellular Specificty of ^177^Lu-DOTATATE and Ligandtracer Experiments

To assess the cellular uptake of ^177^Lu-DOTATATE, a specificity assay was performed. 1 × 10^5^ IMR-32, 7 × 10^4^ N2a, 2 × 10^5^ NB1, 2 × 10^5^ NB2 or 7 × 10^4^ SKNAS cells were seeded in 24-well plates. For NB1 and NB2, the assay was run with suspension cultures. Cells were passaged 24–48 h prior to addition of ^177^Lu-DOTATATE. For the other cell lines, the assay was performed on monolayer cultures where cells were seeded and incubated overnight. ^177^Lu-DOTATATE was added with a final concentration of 10 nM. Blocked control wells with 10 nM ^177^Lu-DOTATATE and 1 μM unlabeled DOTATATE were included as well, where unlabeled DOTATATE was added 5–10 min before the labeled compound. Total volume in each well was 1 mL for adherent cells and 2 mL for PDX cells. Plates were incubated at 37 °C for an additional 24 h. Unbound ^177^Lu-DOTATATE was removed and cell-associated radioactivity was quantified in a gamma counter (1480 Wizard 3”, Wallace). Differences between unblocked and blocked groups were analyzed with unpaired t-tests.

To further validate the specificity assay and assess internalization patterns, the uptake of ^177^Lu-DOTATATE on IMR-32 cells was measured with Ligandtracer (Ridgeview Instruments, Uppsala, Sweden). 2 × 10^6^ IMR-32 cells were seeded on a tilted Petri dish and incubated at 37 °C for 2 days, with media renewal after 1 day, before mounting in a LigandTracer Yellow. The real-time uptake of ^177^Lu-DOTATATE on the cells was measured and corrected to a cell-free reference area on the dish. The assay was performed at room temperature or 37 °C with stepwise addition of 0.3 and 1 nM ^177^Lu-DOTATATE to study association. For the dissociation phase, media was then removed and replaced with fresh media without ^177^Lu-DOTATATE. For assays at room temperature, CO_2_-independent media (Gibco/Thermo Fisher, Waltham, MA, USA) was used to avoid cell detachment. The data was fitted to a kinetic 1:1 (room temperature) or 1:2 (37 °C) model using TraceDrawer 1.9 (Ridgeview Instruments, Uppsala, Sweden).

### 2.8. Multicellular Tumor Spheroids

In order to study the long-term therapeutic response of ^177^Lu-DOTATATE and VIP116, multicellular tumor spheroids were used. This method is specifically beneficial for studies of targeted radionuclide therapy, as the energy from the radionuclide spreads in a three-dimensional space, and the oxygen gradient in the spheroid, which greatly affects the response to radiation, is more similar to an in vivo setting compared to monolayer cultures [43,44].

Agarose-coated 96-well plates were prepared according to the protocol by Friedrich et al. [44]. 4000 IMR-32, 2000 NB1 or 4000 NB2 cells were seeded in 200 μL cell media and incubated at 37 °C until spheroids formed (3–5 days). Six spheroids/group were then treated with ^177^Lu-DOTATATE (0.03–10 kBq/well), VIP116 (0.5–32 μM) or the combination of the two. For combination treatment, VIP116 was added 3 h after ^177^Lu-DOTATATE. Media renewal (100 μL out, 100 μL in) was done at day 0 and 8 and 50 μL fresh media was added at day 3. Spheroids were followed by photography every 2–4 days using a Canon EOS 700D (Canon, Tochigi, Japan) mounted on a Nikon Diaphot-TMD microscope (Nikon, Tokyo, Japan). The cross-section area was measured using Fiji and the volume of each spheroid was calculated, assuming a spherical shape [45]. Comparison between groups were performed on data at day 12 (NB2) or 14 (IMR-32, NB1) using one-way ANOVA with multiple comparisons. IC_50_ was calculated by fitting the data to a dose-response curve (variable slope-normalized response).

As additional controls, spheroids were also treated with DMSO (solvent of VIP116), unlabeled DOTATATE and free ^177^Lu (chelated to DOTA, but without a binding moiety) at concentrations corresponding to the highest doses of each compound. Spheroids were then treated in the same manner as described above.

Bliss and Zero interaction potency (ZIP) models were used to assess synergy between ^177^Lu-DOTATATE and VIP116. Calculations were performed with SynergyFinder 2.0 [46]. 25 combinations within the therapeutic window of each compound (approx. IC_10_-IC_90_) was tested. At least 5 single doses of each compound were included in each assay in order to fit a dose-response curve for the ZIP model. Wilcoxon signed rank test was used to analyze if the mean synergy score was significantly different from 0 (corresponding to an additive effect).

### 2.9. Cell Lysate Preparation and Western Blot

Western blot was used to analyze SSTR2 expression and molecular effects of the treatments. For SSTR2 expression, untreated IMR-32, NB1 and NB2 spheroids and SKNAS monolayer cells were washed with cold PBS. IP lysis buffer (Thermo Fisher, Sweden) with protease and phosphatase inhibitor cocktail (Thermo Fisher) was added to cells and incubated for 30 min on ice, with subsequent centrifugation. Supernatant was collected and kept at −20 °C until further use. Protein concentration was assessed using BCA analysis (Pierce BCA Protein Assay, Thermo Fisher), with subsequent quantification in an Infinite M200 Pro plate reader (Tecan, Zürich, Switzerland).

For analysis of treatment effects, IMR-32 and NB2 lysates were prepared from spheroid cultures. As NB2 grows in neurospheres, cells were passaged as normal and treatment was initiated 6 days after passaging. For IMR-32, 6 million cells per flask were seeded in 36 mL cell media in non-treated 600 mL tissue culture flasks (VWR, Spånga, Sweden). Flasks were swirled 24–48 h after seeding to avoid cell attachment and spheroids were allowed to form for 7 days prior to starting treatment.

At treatment start, spheroids were spun down gently and media was renewed. ^177^Lu-DOTATATE and/or VIP116 were added to cells and flasks were incubated for 1–96 h. The activity concentration of ^177^Lu-DOTATATE was calculated from the spheroid experiments. For IMR-32, an activity concentration of 0.5 Bq/μL was used which corresponded to 0.1 kBq/well in the spheroid experiments. For NB2, 1.5 Bq/μL corresponding to 0.3 kBq/well was used. In combination groups, VIP116 was added 3 h after ^177^Lu-DOTATATE. Lysates were prepared as described above.

For western blot, all components and reagents are listed in Appendix A. Cell lysate was mixed with loading buffer and reducing agent and heated at 70 °C for 10 min prior to loading on a 4–12 % Bis-Tris gel with MOPS running buffer. A total protein amount of 26 (SSTR2) or 15 (all except SSTR2) μg per well was used. Gels were run at 125 V for 70 min and subsequently transferred to ethanol-activated PVDF membranes at 22 V for 40 min. Membranes were blocked for 60 min prior to incubation of primary antibodies against SSTR2 (Invitrogen/Thermo Fisher PA3109, 1/250 Waltham, MA, USA), p53 (Abcam ab32389, 1/1000 Cambridge, UK), MDM2 (Abcam ab259265, 1/500), MYCN (Abcam ab16898, 1/250), p21 (Abcam ab109520, 1/1000), Bax (Abcam ab32503, 1/1000), cleaved caspase 3 (Abcam ab13847, 1/500), E-cadherin (Abcam ab40772 1/10,000) and β-actin (Sigma Aldrich A5441, 1/5000) overnight at 4 °C. Membranes were washed 3 times in PBS with 0.1% Tween-20 for 5 min before incubating with horseradish peroxidase-labeled secondary antibodies against mouse (Invitrogen 626520, 1/10,000) and rabbit (Invitrogen 656120, 1/10,000) in PBS-Tween with 1% BSA for 1 h at room temperature. Membranes were washed 3 times with PBS-Tween before developing immunoreactive bands with ECL solution. Bands were visualized with an Amersham ImageQuant 800FL imager (Cytiva Life Science, Uppsala, Sweden). Image quantification was done with Fiji [45]. All bands were normalized against the β-actin loading control and the untreated control for the corresponding time point.

### 2.10. Flow Cytometry

To assess the cell cycle distribution after treatment, flow cytometry was performed. Spheroid culture flasks were prepared and treated as described for cell lysate preparations. At 24–96 h after treatment, spheroids were gently spun down and washed once with PBS. Single cell suspensions were prepared using trypsin (IMR-32) or accutase (NB2), washed once with PBS and resuspended in PBS. Cells were counted and fixed by adding ethanol to a final concentration of 85%. Samples were kept at –20 °C for a minimum of one week to ensure cell permeabilization.

For staining, cells were washed using PBS with 1% BSA and resuspended in PBS with 1% BSA and 0.1% Triton X-100 at a cell concentration of 1 × 10^6^/mL. After 30 min incubation at room temperature, Fx Cycle Violet (Thermo Fisher, Waltham, MA, USA) was added to a concentration of 1 µg/mL and incubated at 4 °C for 30 min protected from light. Cell cycle distribution data was collected with a Cytoflex S (Beckman Coulter, Indianapolis, IN, USA) and analyzed with FlowJo 10 (FlowJo/BD, Ashland, OR, USA). A minimum of 1 × 10^5^ single cell events for each sample were included in the analysis.

## 3. Results

### 3.1. Labeling

^177^Lu-DOTATATE was successfully prepared with a minimum radiochemical yield of 97% according to ITLC and HPLC (Appendix A). Moreover, stability assays in serum and EDTA solutions exhibited less than 6% dissociated ^177^Lu after 96 h of incubation (Appendix A).

### 3.2. Characterization of Cell Lines

Initial characterization of the neuroblastoma cell lines included short-term response to VIP116 assessed through XTT viability assays and sequencing of the p53 gene. NB1, NB2 and IMR-32 cell viability was inhibited by VIP116 (Figure 1A). SKNAS and N2a were less responsive, where a concentration of 40 μM VIP116 was not able induce a 50% reduction of cell viability. IC_50_ values for all cell lines are listed in Table 1. The response to VIP116 correlated well with the p53 status (Table 1, Appendix A). DMSO did not have an effect on cell viability (Appendix A). The p53-specific effects of VIP116 were further validated in HCT116 (wtp53) and HCT116 p53-/- cells (Appendix A).

To assess the DOTATATE uptake, an uptake and specificity assay was performed (Figure 1B). All cell lines except SKNAS presented with blockable uptake of ^177^Lu-DOTATATE, with IMR-32 demonstrating the highest uptake of 0.33 (95% CI 0.28-0.37) pmol/10^5^ cells. The expression of human SSTR2 was further validated with Western blot, where all cell lines exhibited detectable SSTR2 expression (Appendix A).

The binding kinetics of ^177^Lu-DOTATATE binding was further characterized by LigandTracer analyses using IMR-32 as a model system (Figure 1C). At 37 °C, the maximum ^177^Lu-DOTATATE uptake was four-fold higher than at 20 °C. The affinity from the 1:1 model was 0.38 nM at room temperature. For 37 °C, where a 1:2 model was used, the calculated affinities were 0.17 and 8.96 nM.

### 3.3. ^177^Lu-DOTATATE and VIP116 Monotherapy Inhibit Spheroid Growth

To mimic a more in vivo-like setting and study long-term treatment effects, IMR-32, NB1 and NB2 cells were grown as multicellular tumor spheroids and followed for 12–14 days after treatment with ^177^Lu-DOTATATE or VIP116 (Figure 2). Potential unspecific effects of DMSO, unlabeled DOTATATE and ^177^Lu were also assessed (Appendix A–D). IC_50_ values at the endpoint are presented in Table 1.

The spheroid growth was significantly inhibited by ^177^Lu-DOTATATE (Figure 2A,D,G) with complete growth inhibition at 1 kBq for IMR-32 and 10 kBq for NB2 and NB1. Unlabeled DOTATATE did not have an effect on spheroid growth (Appendix A–D). Unspecific ^177^Lu did not have an effect at doses lower than 3 kBq, whereas 10 kBq inhibited NB1 and NB2 spheroid growth slightly (Appendix A–D).

VIP116 treatment correlated well with previous results from monolayer assays, with spheroid growth inhibition in all three cell lines (Figure 2B,E,H). Complete growth inhibition was reached at 32 μM for IMR-32, 4 μM for NB2 and 8 μM for NB1. Moreover, treatment with VIP116 resulted in an increase in p53 protein levels in IMR-32 and NB2 (Figure 2J,K). Maximum p53 fold-change was 2.7 and 3.3 for IMR-32 and NB2, respectively.

### 3.4. Combination Therapy Induces Varying Degree of Synergy in IMR-32 and NB2 Spheroids

In order to explore combination effects between ^177^Lu-DOTATATE and VIP116, IMR-32 and NB2 spheroids were treated with a combination modality and followed over time (Figure 3). A total of 25 dose combinations were chosen within the therapeutic window of the compounds for each cell line. Synergy calculations were performed at the assay endpoint.

For IMR-32, VIP116 was able to strongly potentiate the effects from ^177^Lu-DOTATATE, leading to decreased spheroid growth or even spheroid disintegration at higher doses (Figure 3A,B). The effects were synergistic with mean synergy scores (with 95% CI) of 8.7 (6.0–11.5) and 8.6 (6.2–11.0) for Bliss and ZIP models, respectively (Figure 3C). Individual size ratios and synergy values for each combination are presented in Appendix A.

VIP116 was also able to potentiate ^177^Lu-DOTATATE in NB2 spheroids (Figure 3D,E), although not to such large degrees as seen in IMR-32. Although demonstrating the same trends in treatment effect, the intensity of the effects varied between repeated experiments (Appendix A), which complicated synergy assessments. Thus, although individual experiments displayed synergy, the mean synergy values, 1.5 (−1.4–4.5) and 1.7 (−0.8–4.2) from Bliss and ZIP models, respectively, indicated an additive effect (Figure 3F).

The p53-specific radiosensitizing effect of VIP116 was further validated in HCT116 with wtp53 and HCT116 p53-/- cells (Appendix A).

### 3.5. Combination Treatment Activates p53 Pathway with Differential Effects on MYCN and MDM2

To further characterize the combination therapy and explain synergy effects, protein expression of p53 and downstream targets involved in cell cycle (p21), apoptosis (Bax, cleaved caspase 3) and regulation of p53 (MYCN, MDM2) was determined with western blot 6–96 h after treatment of IMR-32 and NB2 spheroids (Figure 4, Appendix A).

VIP116 and combination therapy increased p53 levels in both cell lines. In IMR-32 p53 was induced at 6 h, and levels were consistently higher than control at all tested time points. The p53 levels in the combination group demonstrated higher levels than VIP116 group at 6 and 24 h. For NB2, the p53 levels followed a similar pattern with increased expression as a result of VIP116 or combination therapy and the strongest induction at earlier time points. ^177^Lu-DOTATATE demonstrated very little effect on the p53 expression.

In IMR-32, VIP116 and combination therapy induced p21 with a maximum fold change of 11.4 at 48 h for VIP16 and 8.6 at 24 h for combination therapy. In NB2, p21 induction was visible but not as strong, with a maximum fold change of 2.3 for VIP116 and 3.8 for combination therapy, both at 24 h. Similar to p53, ^177^Lu-DOTATATE demonstrated very little effect on the p21 expression. As p21 is an inhibitor of cell cycle progression at G1/S and G2/M, the cell cycle distribution at 24–96 h was determined with flow cytometry. No major differences in cell cycle distribution were detected for either cell line (Appendix A).

Both cell lines displayed increased Bax levels after treatment with ^177^Lu-DOTATATE, VIP116 and combination treatment, although the strongest effects were seen in VIP116 and combination groups. Similar to Bax, there was an induction of cleaved caspase 3 in both IMR-32 and NB2 after treatment. Moreover, the levels of cell-to-cell adhesion molecule E-cadherin exhibited a slight increase from 24 h and forward in all treated groups.

^177^Lu-DOTATATE demonstrated an inhibitory effect of MYCN protein expression in IMR-32, especially at 72 and 96 h. VIP116 and combination treatment followed the same pattern, although not as pronounced. In contrast, MYCN expression increased in NB2 after VIP116 and combination therapy at earlier time points, while ^177^Lu-DOTATATE demonstrated very little effect on the MYCN expression.

In IMR-32, MDM2 was induced by VIP116 and combination therapy at 6 h, but then declined to baseline levels. The levels of MDM2 after ^177^Lu-DOTATATE treatment were similar to or slightly lower than control. In NB2, expression of MDM2 followed a different pattern, with elevated MDM2 levels in the VIP116 and combination groups. The strongest induction was seen in the combination therapy group with a fold-change of 14.2 at 24 h. MDM2 levels in the ^177^Lu-DOTATATE group fluctuated through the investigated time points, with a fold-change between 0.4 and 1.3.

## 4. Discussion

In the current study, the novel MDM2/MDM4-p53 antagonist VIP116 was explored as means to potentiate targeted radionuclide therapy with ^177^Lu-DOTATATE in neuroblastoma. This heterogenous group of pediatric tumors include a subset of high-risk patients who are in need of more therapeutic alternatives.

The mutation rate of p53 is reported to be as low as 2 % in neuroblastoma [52], making VIP116 a suitable drug candidate in this patient group, as it requires wtp53 to be effective. Furthermore, overexpression of SSTR2, one of the receptor targets for ^177^Lu-DOTATATE, is found in 75–90 % of all neuroblastoma cases [20,21,22,23,24]. Consequently, there is strong clinical relevance to combine ^177^Lu-DOTATATE with VIP116 in neuroblastoma. Furthermore, the use of combination therapy may overcome or prevent drug resistance that may occur from either monotherapy, such as p53 mutations or heterogeneous SSTR density [5,18,53].

The study was performed both in commercially available cell lines and by the use of patient-derived xenograft (PDX)-models established from neuroblastoma patients. PDX cell lines have previously been demonstrated to recapitulate the hallmarks of high-risk neuroblastoma, retain the genotype and phenotype of patient tumors, and serve as clinically relevant models for studying and targeting high-risk metastatic neuroblastoma [38].

The initial characterization (Figure 1A, Table 1) concluded that all p53wt cell lines responded well to VIP116 treatment, with micromolar IC_50_ values. SKNAS did not produce PCR products for exon 10 and 11, which was further supported by previous studies, and did not reach 50% inhibition by 40 μM VIP116 [35,50]. The previously uncharacterized cell line N2a presented with a missense mutation in exon 5 (Table 1, Appendix A). It has not been proven whether this mutation has an effect on p53 function, but based on the lack of response to VIP116, we speculate that this might be the case.

Cellular uptake and specificity studies (Figure 1B,C) validated the SSTR-status of the cells and specific binding of the conjugate. IMR-32 xenografts have previously been reported to be SSTR positive, which is in line with our results [20]. However, SKNAS, which did not have any detectable uptake of ^177^Lu-DOTATATE (Figure 1B) but expression of SSTR2 (Appendix A), was reported to be SSTR positive in the same study. The discrepancies may be due to the choice of assay or model system. We conclude that SKNAS have no or very low uptake of ^177^Lu-DOTATATE in our experimental settings. These experiments confirmed the prerequisites for the study, leading to further investigation of the wtp53 and SSTR expressing cell lines IMR-32, NB1 and NB2.

The binding and internalization of ^177^Lu-DOTATATE was further characterized on the strongly SSTR positive cell line IMR-32. The real-time uptake of ^177^Lu-DOTATATE was assessed in both 37 °C and room temperature, as lower temperatures inhibit internalization. The uptake was four-fold lower at room temperature, indicating a high internalization rate. These results are in line with previous studies on SSTR ligand internalization [54,55]. Additional studies are currently being performed to characterize the interaction further.

The effects of VIP116 or ^177^Lu-DOTATATE as monotherapies were then assessed in multicellular tumor spheroids (Figure 2). Although spheroid assays generally have a lower throughput and are more laborious than monolayer assays, these 3D models serve as a more in vivo-like environment [43,44]. It is especially suitable for assessing effects of molecular radiotherapy, as cross-dose radiation from neighboring cells and late radiotherapy effects can be studied.

The multicellular spheroid monotherapy assays correlated well with the monolayer characterizations, and induced dose-dependent spheroid growth inhibition in all three models. This further validates the potential of p53 stabilization and ^177^Lu-DOTATATE therapy in neuroblastoma. Moreover, the fact that spheroids from both the immortalized cell line and the PDX-derived neuroblastoma cells responded well to treatments is promising.

To further validate that VIP116 mediated an increase of p53 levels, VIP-116 treated IMR-32 and NB2 lysates were stained for p53 expression 1 h to three days after treatment. As expected, the expression of p53 increased in both cell lines after treatment. This is in line with previous experiences with VIP116 and its predecessor PM2 [10,12,14].

Spheroids were then treated with a combination of ^177^Lu-DOTATATE and VIP116 and the degree of synergy was assessed with Bliss and ZIP synergy models (Figure 3). Both IMR-32 and NB2 responded well to combination therapy, where the resulting spheroid volume was clearly reduced compared to corresponding monotherapies. For example, 0.3 kBq ^177^Lu-DOTATATE together with 16 μM VIP116 completely suppressed IMR-32 spheroid growth (Figure 3A). This is in contrast to the group that received 0.3 kBq ^177^Lu-DOTATATE as a monotherapy, where spheroid growth was resumed after approximately 10 days. 16 μM VIP116 caused a minor growth delay, but spheroid size did not significantly differ from control group at endpoint. For ^177^Lu-DOTATATE monotherapy, a dose of 1 kBq was required to achieve complete growth inhibition (Figure 2A). Although results cannot be directly translated to a clinical setting, these demonstrated effects are encouraging. If the radiation dose could be reduced by a factor of 3 in a clinical setting, it could potentially mean a reduction of side-effects without risking therapeutic response. The reduction of side effects is especially appealing when treating pediatric patients, as long-term side effects from the therapy can greatly affect the quality-of-life throughout the patient’s life. These results are in line with our previous experiences using p53-activating stapled peptides as potentiators of radiotherapy [12,13,14].

The spheroid volume at the assay endpoint was used as basis for synergy calculations. In IMR-32, a significant synergy between ^177^Lu-DOTATATE and VIP116 was observed. In NB2, the degree of synergy varied greatly between different combinations, with a mean synergy value indicating an additive effect. These results strengthened the rationale for combination treatment and highlighted the importance of studying drug synergy in several models.

To further understand the synergy and the treatments effects on p53 signaling, protein expression of p53 and downstream targets were analyzed (Figure 4). As expected, p53 was upregulated by VIP116 and combination therapy. However, effects from ^177^Lu-DOTATATE on p53 expression were small. This is in contrast to previous studies with external beam radiation, where p53 expression increased after treatment [14,47]. One possible explanation can be the different dose rates of external beam radiation, where the radiation event is limited to minutes, compared to targeted radionuclide therapy where radiation is continuously delivered during hours or even days. The rate of DNA double strand break (DSB) induction is indeed very different between external beam radiation and ^177^Lu-DOTATATE, previously demonstrated by O’Neill et al. [56]. It is possible that the amount of DSBs caused by ^177^Lu-DOTATATE at the studied time points was not sufficient to induce a detectable increase of p53 protein expression in these settings. Nevertheless, the induction of p53 was higher in the combination group than VIP116 monotherapy group 6 h after treatment for both IMR-32 and NB2. This indicates that a stronger p53 induction may mediate the growth inhibitory effects.

There are several growth inhibitory processes than can occur after treatment. We first looked into cell cycle arrest. The CDK inhibitor p21 is a transcriptional target of p53 and can inhibit cell cycle progression at G1/S and G2/M [57]. Despite induction of p21 (Figure 4) in both IMR-32 and NB2, no difference in cell cycle distribution was observed (Appendix A), in contrast to our earlier experiences with p53-stabilizing stapled peptides [14]. This may be a result of MYCN-mediated cell cycle progression, which is further discussed below.

We then investigated activation of apoptosis through Bax and cleaved caspase 3. Results confirmed both cell lines activate apoptosis after both mono- and combination treatment (Figure 4). The induction of Bax, which is transcriptional target of p53, further proves that the apoptosis is p53-dependent. Additionally, analysis of the MET-marker E-cadherin in IMR-32 spheroids was done to investigate the metastatic potential of the cells. E-cadherin was upregulated 24–96 h after treatment, indicating that treatment response was not associated with increased metastatic potential (Figure 4A).

Apart from cell cycle arrest and apoptosis, the spheroid growth may be influenced by additional pathways which have not been addressed in the current study. As neither of the cell lines studied here arrest the cell cycle after radiotherapy, mitotic catastrophe may occur if there is insufficient time to repair the DNA damage prior to entering mitosis. This is supported by a previous study, which studied mitotic catastrophe after inhibition of DNA repair in neuroblastoma [48]. Furthermore, p53 and subsequently p21 can induce senescence, which is defined as permanent withdrawal from the cell cycle leading to an increase in the G1 population [58]. Although we saw induction of p21, this did not affect the cell cycle distribution as previously mentioned.

p53 signaling is highly influenced by MYCN, an oncogenic transcription factor amplified in approximately 20% of all neuroblastomas. It is a prognostic marker for poor survival [59]. As mentioned above, the lack of p21-mediated cell cycle arrest may be explained by MYCN activity. A previous study has concluded that MYCN-amplified neuroblastoma cells have defective G1-arrest despite induction of p21 [47]. In the current study, both IMR-32 and NB2 were MYCN amplified, which can explain the lack of cell cycle arrest.

Additionally, as MYCN induces transcription of the p53 negative regulator MDM2, it can greatly affect p53 signaling [60]. IMR-32 exhibited reduced levels of MYCN after treatment, especially in the ^177^Lu-DOTATATE group. In contrast, ^177^Lu-DOTATATE mediated a meager effect on MYCN expression in the NB2 model. Furthermore, NB2 displayed increased levels of MDM2, with a maximum fold-change of 14.2 in the combination group (Figure 4). IMR-32 demonstrated a much weaker induction of MDM2, with a maximum fold-change of 1.6 in combination group. This may be explained by the fact that MDM2 is a transcriptional target of MYCN, and lower MYCN levels results in lower MDM2 induction. As increased MYCN and by extension MDM2 activity can suppress p53-mediated growth inhibition, this may explain the differences in synergy between IMR-32 and NB2. However, the interplay between p53 and MYCN is complex as both transcription factors have a large number of downstream targets, and warrants further investigations.

## 5. Conclusions

The present study demonstrates that p53 stabilization with VIP116, as well as targeted radionuclide therapy with ^177^Lu-DOTATATE, are feasible treatment options for neuroblastomas. Moreover, the combination of VIP116 and ^177^Lu-DOTATATE is particularly promising, due to synergistic effects. The synergy may be influenced by MYCN activity and it is important to further characterize the interplay between p53 and MYCN in the context of radiotherapy (Figure 5).

The combination of VIP116 and ^177^Lu-DOTATATE may improve the treatment efficacy of patients with advanced stage neuroblastoma, leading to reduced side effects and increased cure rates. Further in vivo studies investigating this strategy are required to confirm these promising findings.

## Figures and Tables

**Figure 1 biomolecules-11-01695-f001:**
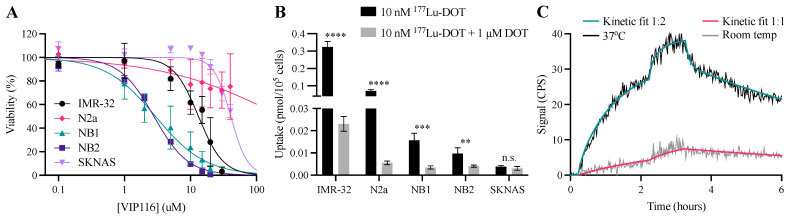
Characterization of neuroblastoma cell lines. (**A**) XTT viability assay after 72 h incubation with 0.1–40 μM VIP116. IC_50_ values are listed in Table 1. Pooled data from four independent experiments (mean ± SD). (**B**) Cellular specificity of ^177^Lu-DOTATATE. Uptake of ^177^Lu-DOTATATE without (black bars) or with (grey bars) blocking. Data from one representative experiment (mean ± SD, *n* = 4). n.s. = not significant, ** = *p* < 0.01, *** = *p* < 0.001 and **** = *p* < 0.0001. (**C**) Real-time uptake of ^177^Lu-DOTATATE in IMR-32 cells in room temperature (grey) or at 37 °C (black) with 1:1 (red) or 1:2 (green) kinetic models. Data from one representative experiment.

**Figure 2 biomolecules-11-01695-f002:**
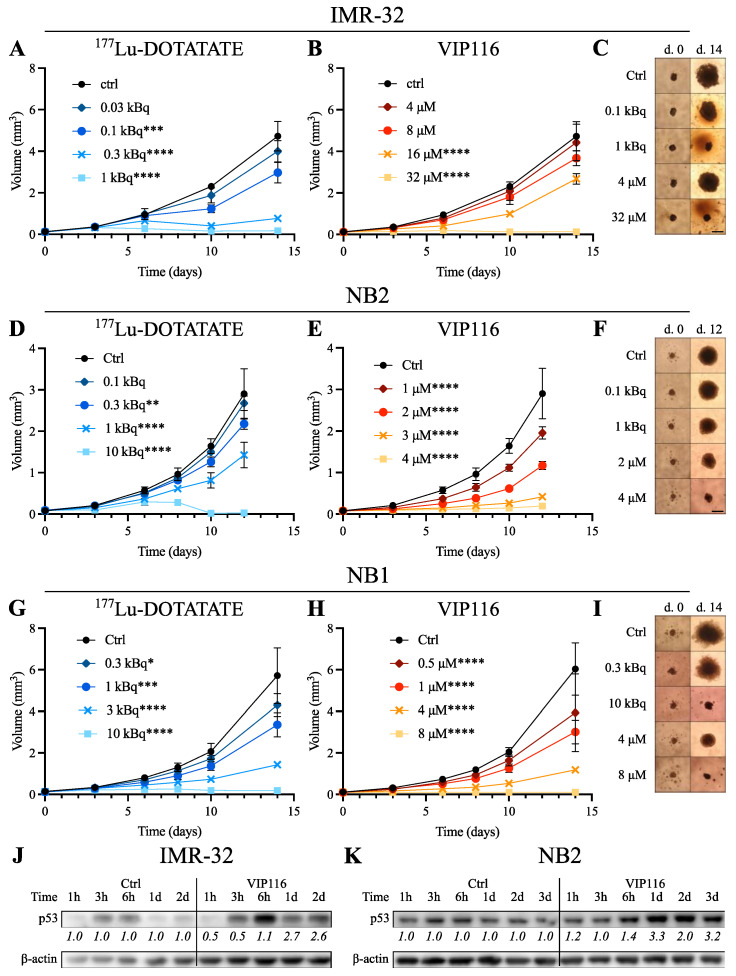
Spheroid monotherapy. Graphs from one representative experiment (mean±SD, *n* ≥4). Scale bar on spheroid images = 500 μm. (**A**) ^177^Lu-DOTATATE monotherapy of IMR-32. (**B**) VIP116 monotherapy of IMR-32. (**C**) Representative IMR-32 spheroid images at days 0 and 14. (**D**) ^177^Lu-DOTATATE monotherapy of NB2. (E) VIP116 monotherapy of NB2. (**F**) Representative NB2 spheroid images at days 0 and 12. (**G**) ^177^Lu-DOTATATE monotherapy of NB1. (**H**) VIP116 monotherapy of NB1. (**I**) Representative NB1 spheroid images at day 0 and 14. (**J**) Representative Western blot of VIP116-treated IMR-32 spheroids. Intensity value for p53 were normalized to β-actin and corresponding control band for each time point. (**K**) Representative Western blot of VIP116-treated NB2 spheroids. Intensity value for p53 were normalized to β-actin and corresponding control band for each time point. n.s. = not significant, * = *p* < 0.05, ** = *p* < 0.01, *** = *p* < 0.001 and **** = *p* < 0.0001.

**Figure 3 biomolecules-11-01695-f003:**
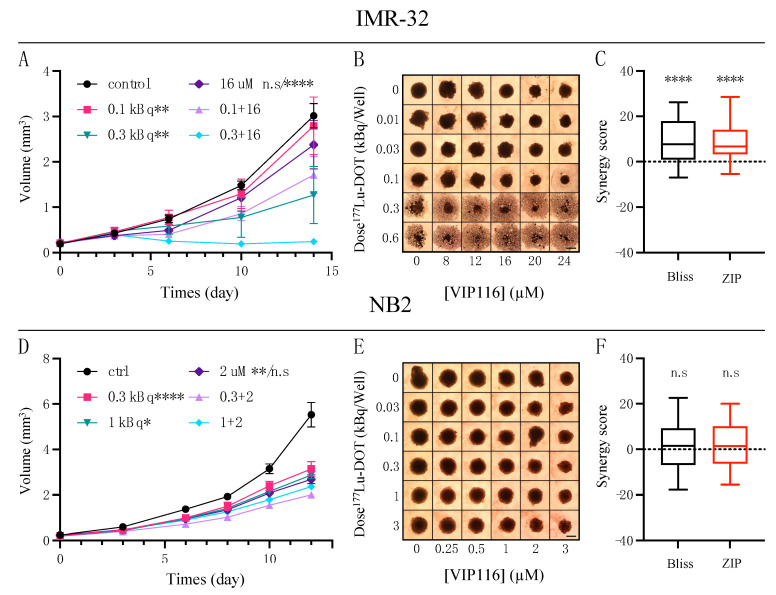
Spheroid combination therapy. (**A**) Representative graph (mean ± SD, *n* ≥ 4) of IM R-32 spheroids treated with 0.1 or 0.3 kBq ^177^Lu-DOTATATE and 16 μM VIP116. *p*-value indications next to monotherapy legends are in comparison to the corresponding combination group(s). (**B**) Representative IM R-32 spheroid images at day 14. Scale bar = 500 μm. (**C**) Synergy values for IM R-32 spheroid combinations using Bliss (black) or ZIP (red) synergy models. Values below 0 indicate antagonism, 0 indicates additive effect and above 0 indicates synergy. Pooled data from two independent experiments (n = 50) are presented as mean with box (25th to 75th percentile) and whiskers (min to max). (**D**) Representative graph (mean ± SD, *n* ≥ 4) of NB2 spheroids treated with 0.3 or 1 kBq ^177^Lu-DOTATATE and 2 μM VIP116. *p*-value indications next to monotherapy legends are in comparison to the corresponding combination group(s). (**E**) Representative NB2 spheroid images at day 12. Scale bar = 500 μm. (**F**) Synergy values for NB2 spheroid combinations using Bliss (black) or ZIP (red) synergy models. Values below 0 indicate antagonism, 0 indicates additive effect and above 0 indicates synergy. Pooled data from two independent experiments (*n* = 50) are presented as mean with box (25th to 75th percentile) and whiskers (min to max). n.s. = not significant, * = *p* < 0.05, ** = *p* < 0.01 and **** = *p* < 0.0001.

**Figure 4 biomolecules-11-01695-f004:**
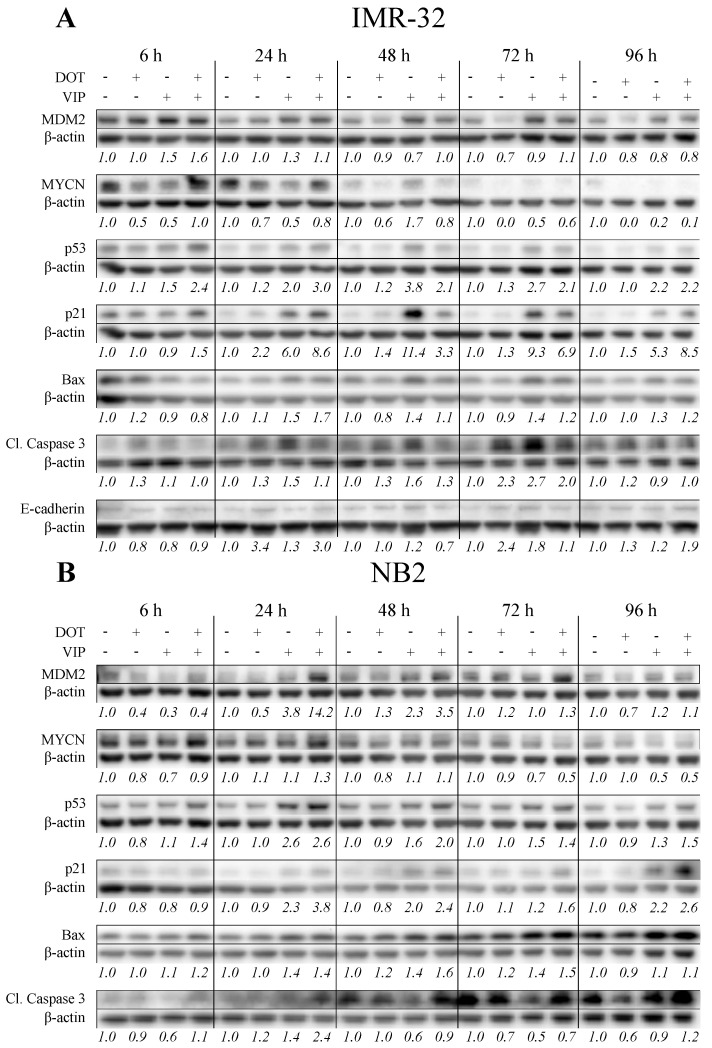
Western blot. Expression of MDM2, MYCN, p53, p21, Bax, cleaved caspase 3 and E-cadherin in (**A**) IMR-32 and (**B**) NB2 6–96 h after treatment with ^177^Lu-DOTATATE and/or VIP116. Representative data from one experiment. Intensity values, shown below each band, was normalized against the loading control (β-actin) and the corresponding untreated control for each time point. Intensity plot from all replicates can be found in Appendix A.

**Figure 5 biomolecules-11-01695-f005:**
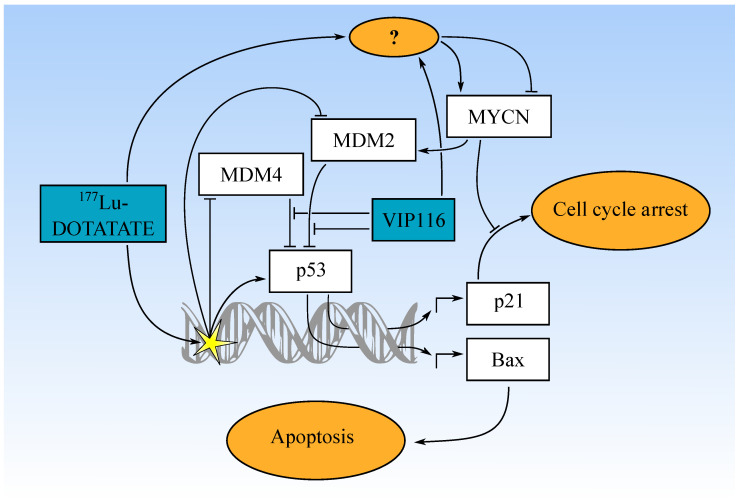
Suggested mechanism of VIP116-mediated potentiation of ^177^Lu-DOTATATE in neuroblastoma. ^177^Lu-DOTATATE causes DNA double strand breaks which initiates a slight p53-mediated response. VIP116 is able to enhance the p53 response and induce transcription of downstream targets, leading to increased apoptosis. MYCN, which can be induced or repressed by the therapies via unknown mechanisms, can affect p53 function negatively via MDM2 transcription or blocking p21-mediated cell cycle arrest.

**Table 1 biomolecules-11-01695-t001:** p53 status, MYCN status and IC_50_ values from XTT and spheroid monotherapy assays. IC_50_ data is presented as mean value (95% CI). wt = wildtype, mut = mutated, n.e. = not established, del = deletion, amp = amplified, n.amp = not amplified, Sph = spheroid assay, Ref = additional references for p53 and MYCN status.

Cell Line	p53 Status	MYCN Status	XTT VIP116 (μM)	Sph. VIP116 (μM)	Sph. ^177^Lu-DOT (kBq/well)	Ref.
IMR-32	wt	amp	13.2 (11.1–15.4)	15.3 (12.7–18.1)	0.12 (0.10–0.14)	[35,47,48]
NB1	wt	amp	2.9 (2.3–3.7)	0.9 (0.8–1.0)	1.1 (0.8–1.5)	[39]
NB2	wt	amp	2.8 (2.6–3.0)	1.4 (1.2–1.6)	0.88 (0.67–1.21)	[38]
N2a	mut ^1^	amp	238.2 (60.5–10^5^) ^3^	n.e.	n.e.	[49]
SKNAS	del ^2^	n.amp	40.8 (27.8–45.4) ^3^	n.e.	n.e.	[35,50,51]

^1^ Missense mutation in the DNA-binding domain of exon 5. Only exon 2 and 4–11 were sequenced. ^2^ No visible PCR product of exon 10 and 11, supported by additional references. ^3^ A total of 50% inhibition was not reached at the maximum concentration.

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
