# Peer review of "p53-Mediated Radiosensitization of 177Lu-DOTATATE in Neuroblastoma Tumor Spheroids"

_biomolecules, 2021, doi:10.3390/biom11111695_

Round 1
Reviewer 1 Report
The current study falls within the scope of previous developed work by the same authors, published last year “Tumor-Targeted Delivery of the p53-Activating Peptide VIP116 with PEG-Stabilized Lipodisks”. The utilization of 3D models has several advantages in comparison to the use of the traditional monolayer cultures, resembling many aspects of the pathophysiological environment of human tumors. The current work seems highly relevant given the current state of the art.
English language and style are fine.
The introduction focus on the main topic, it is effective and well organized. References supporting the Introduction section are accurate and well-founded and number of citations is appropriate. A minor comment:
- Line 166-170: The authors describe the advantages of using a tumor spheroids model. In your opinion are there any negative aspects of using this type of model? Should be interesting to discuss what kind of limitations bring this sort of model and/or compare it to other models.
The results are well described with adequate support of images and graphs. The initial characterization of the cell lines (Figure 1.) gives support to the further results attained. Some minor comments:
- 177Lu-DOTATATE and VIP116 monotherapy inhibit spheroid growth: For the evaluation of 177Lu-DOTATATE and VIP116 monotherapy, was the spheroid volume the only analytical end point used for evaluating these drug testing? In the authors opinion can it be influencing conclusions attained? How this it compare with other methods to evaluate drug efficacy?
- Table 1. Proposed discussion: Can the low IC50 values determined especially in NB1 and NB2 cell lines influence the further results? The question arises since concentrations up to 8µM of VIP116 are used during further experiments (Figure 2.).
In conclusion, it is in my opinion that these are promising preliminary results, despite not being possible to translate them directly into a clinical setting. More studies will be needed and in particular in vivo data will be crucial.
So, congratulations for the good work!
Reviewer 2 Report
Lundsten et al. investigated the therapeutic effect of a VIP116 and 177-Lu-DOTATATE monotherapy and a combinational approach with both drugs in the context of neuroblastoma. Neuroblastoma tumor express SSTR, which is targeted by 177-Lu-DOTATATE. Since neuroblastoma exhibit a low p53 mutation rate and VIP116 stabilizes p53, the combinational therapeutic approach could be beneficial for patients. The study presents a novel finding and is well planned. However, major comments need to be addressed before considering the manuscript for publishing.
Major
- Since NB1 spheroids disintegrate (also IMR-32 spheroids in combinational therapy), did the authors take increased metastatic potential into account? Analyzing mesenchymal-epithelial transition (EMT) marker would give insights into this.
- Why do the different cell lines vary so much in the uptake of 177Lu-DOTATATE (Figure 1B)? Does this correlate with the SSTR expression?
- Does VIP116 also lead in the other p53 positive cell lines to increased p53 protein levels?
- Peak of p53 expression is after 6 hours in IMR-32 cells (Figure 2J). What is the half-life of VIP116? Would the patient in this case need two or more treatments per day to ensure an effective dose?
- The drug dose needed are very different between the cell lines. How will that information be transferred to the patient (e. g. p53 expression analysis before treatment)?
Minor
- Line 32 – 33: the content of the sentence is not connected to the following paragraph (I suggest to delete this sentence)
- Why did the authors choose VIP116 over PM2? Since PM2 is not used in the study, I suggest to exclude PM2 from the manuscript (introduction).
- Please attach original flow cytometry and the original Western blot (with marker) data as supplemental.
- Please indicate the used marker in Figure 4.
- Is anything known about multidrug resistance development following therapy with VIP116, 177Lu-DOTATATE or in combination?
- Please attach the HPLC data (section Results – Labeling).
Reviewer 3 Report
This work investigated the novel p53-stabilizing peptide VIP116 in neuroblastoma, both as monotherapy and together with 177Lu-DOTATATE. Some issues need to be further clarified before further consideration.
- Why authors chose to prepare 177Lu-DOTATATE themselves, but not buy thee commercial 177Lu-DOTATATE. But the authors didn’t show any data about the quality of 177Lu-DOTATATE, which are important for the following experiments.
- Different cell lines showed different 177Lu-DOTATATE cellular uptake efficiency, is this result associated with the somatostatin receptor expression level? Authors should also provide the somatostatin receptor expression level for each cell line.
- the unit Bq is confusing for me, i think it may also confuse other readers, authors should provide more background introduction about it.
- The western blot results seem to be a little strange, as the shape of tested protein bands were not similar with the actin control bands.
- If the radiosensitization effects is p53 dependent, authors should also try to test the effects on p53 knockout or p53 knockdown cell lines to further prove it. This is the most important results to clarify that it’s p53 mediated.
- A mechanism diagram should also be provided to summarize the results.
Round 2
Reviewer 2 Report
The authors addressed all of my comments.
Reviewer 3 Report
Authors did very good reversion following the suggestions